# Inherited Thrombocytopenia Caused by Variants in Crucial Genes for Glycosylation

**DOI:** 10.3390/ijms24065109

**Published:** 2023-03-07

**Authors:** Ana Marín-Quílez, Lorena Díaz-Ajenjo, Christian A. Di Buduo, Ana Zamora-Cánovas, María Luisa Lozano, Rocío Benito, José Ramón González-Porras, Alessandra Balduini, José Rivera, José María Bastida

**Affiliations:** 1Servicio de Hematología y Oncología Médica, Hospital Universitario Morales Meseguer, Centro Regional de Hemodonación, Universidad de Murcia, IMIB-Pascual Parrilla, CIBERER-U765, 30003 Murcia, Spain; 2IBSAL, CIC, IBMCC, Universidad de Salamanca-CSIC, 37007 Salamanca, Spain; 3Department of Molecular Medicine, University of Pavia, 27100 Pavia, Italy; 4Department of Hematology, Complejo Asistencial Universitario de Salamanca (CAUSA), Instituto de Investigación Biomédica de Salamanca (IBSAL), Universidad de Salamanca (USAL), 37007 Salamanca, Spain; 5Department of Biomedical Engineering, Tufts University, Medford, MA 02155, USA

**Keywords:** glycosylation, inherited thrombocytopenia, inherited platelet disorder, syndromic manifestation, thrombopoiesis, platelet clearance

## Abstract

Protein glycosylation, including sialylation, involves complex and frequent post-translational modifications, which play a critical role in different biological processes. The conjugation of carbohydrate residues to specific molecules and receptors is critical for normal hematopoiesis, as it favors the proliferation and clearance of hematopoietic precursors. Through this mechanism, the circulating platelet count is controlled by the appropriate platelet production by megakaryocytes, and the kinetics of platelet clearance. Platelets have a half-life in blood ranging from 8 to 11 days, after which they lose the final sialic acid and are recognized by receptors in the liver and eliminated from the bloodstream. This favors the transduction of thrombopoietin, which induces megakaryopoiesis to produce new platelets. More than two hundred enzymes are responsible for proper glycosylation and sialylation. In recent years, novel disorders of glycosylation caused by molecular variants in multiple genes have been described. The phenotype of the patients with genetic alterations in *GNE, SLC35A1, GALE* and *B4GALT* is consistent with syndromic manifestations, severe inherited thrombocytopenia, and hemorrhagic complications.

## 1. Introduction

Glycosylation is a key process by which carbohydrates or saccharides bind to proteins, lipids, and other biomolecules. It is a highly prevalent, conserved, and complex post-translational alteration [1]. Glycosylation influences a wide range of cellular processes, including control of protein secretion and degradation, cell signaling, adhesion and migration, host–pathogen interactions, or immune defense including both innate and acquired immunity [2,3,4,5].

Glycosylation is a highly modular process, whereby carbohydrate building blocks are repeatedly linked and assembled in varying lengths and branches. It is an unplanned process that gives rise to a wide and diverse repertoire of functional molecules [6] and plays a crucial role in the correct folding of proteins, their stability, and the formation of mature and functional proteins [7].

The presentation of glycans on cell surfaces is governed by more than 200 glycosyltransferases, sugar–nucleotide synthesis, and transport proteins, mainly located in the endoplasmic reticulum and Golgi apparatus [8,9]. The glycoconjugate forms are generally based on nine monosaccharides (Figure 1A). The glycan residues can be conjugated to asparagine (N-glycan) or serine/threonine (O-glycan) residues to form the glycoproteins. Two N-acetylglucosamine (GlcNAc) and three mannose (Man) residues usually constitute the core of N-glycans, which are generally highly branched. We can distinguish high mannose, hybrid, and complex N-linked glycans [10,11]. In contrast, O-glycans are linked with N-acetylgalactosamine (GalNAc) and are, in general, less branched than N-glycans (Figure 1B). Between the N- and O-branches, traces of galactose (Gal), GalNAc, GlcNAc, fucose (Fuc) and the final sialic acid (Sial) can be detected (Figure 1B). Glycosphingolipids are conjugated at the plasma membrane, whereas glycosaminoglycans are mainly composed of an initial xylose (Xyl) followed by glucuronic acid (GlcA) and GlcNAc or GalNAc branches (Figure 1B) [10].

More than half of the proteins in human cells and 50–70% of serum proteins are glycosylated [12]. Platelets express highly glycosylated proteins on their surface, which are involved in platelet hemostasis and function, as well as in their interaction with other cells [13]. To maintain a normal circulating platelet count between 150–400 × 10^9^/L, about 10^11^ of them are cleared daily, highlighting the importance of the balance between production and removal of these cells. Glycosyltransferases and synthesis and transport proteins are involved in both processes, and their dysregulation leads to variations in platelet counts and/or functional alterations [14]. In this review, we will focus on the role of glycosylation for proper platelet formation and clearance, and on the genes involved in platelet physiology whose molecular alterations are associated with inherited thrombocytopenia (IT). 

## 2. Role of Glycosylation in Thrombopoiesis and Platelet Clearance

Thrombopoietin (TPO) is a hematopoietic growth factor essential for thrombopoiesis that is produced predominantly by the liver [15]. Binding of TPO to the c-Mpl receptor (encoded by *MPL*) on platelets and megakaryocytes (MKs) activates a cascade of signaling molecules driving MK development and platelet formation [16]. The plasma concentration of TPO correlates inversely with platelet number, and circulating levels are determined as a function of its binding to platelets and MK, leading to its internalization and degradation along with the c-Mpl receptor [17]. Decreased platelet turnover rate or reduced platelet number results in increased levels of free TPO, which induces a compensatory response dependent on bone marrow MK concentration to increase platelet production [18].

N-linked and O-linked glycans play an essential role in the stability of major MK and platelet surface glycoproteins, including the GPIb-IX-V complex, GPIIb-IIIa (integrin αIIbβ3) and GPVI. Alteration of their glycosylation negatively influences glycoprotein functions, leading to abnormal morphology, defective platelet activation and excessive bleeding [19]. In addition, platelet GPIbα is responsible for the maintenance of steady-state hepatic TPO production [20]. It has been described that the absence of GPIbα in the MK membrane leads to reduced thrombopoiesis due to aberrant membrane development during MK maturation, impaired formation of the membrane demarcation system (DMS), and disruption of the microtubule cytoskeleton, as described in Bernard Soulier syndrome (BSS) [21]. In addition, our group has recently described a defect in GPIbα glycosylation that affects thrombopoiesis and actin cytoskeleton remodeling [22]. These findings highlight the essential role of protein glycosylation during megakaryopoiesis and thrombopoiesis.

Desialylated and/or senescent platelets increase TPO production. Loss of the final sialic acid is responsible for platelet clearance, as exposure of the penultimate Gal residue is recognized by hepatic Ashwell-Morell receptors (AMR), whereas exposure of the GlcNAc residue is recognized by resident hepatic macrophages (Kupffer cells), via the αMβ2 integrin. Consequently, AMR activation drives hepatic TPO mRNA expression through Janus kinase 2 (JAK2) and signal transducer and activator of transcription 3 (STAT3) signaling, triggering a feedback mechanism to increase TPO levels and promote platelet formation [23,24] (Figure 2A). AMR preferentially binds to complex branched glycans, suggesting that N-glycans are the main site of ligand recognition [25]. However, desialylation of O-glycans on GPIbα is known to favor receptor signaling and surface expression of neuraminidase which, by desialylating platelet N-glycans, would allow AMR-mediated clearance. On the other hand, Kupffer cells also play an important role in the clearance of aged platelets and during immune-mediated thrombocytopenia [20,26].

Platelet clearance by aging (senescence) induces signals including loss of sialic acid mediated by up-regulation of platelet sialidases Neu1 and Neu3, which are expressed in the granular and plasma membrane compartments, respectively [27]. Neu1 and Neu3 usually impact sialic acid binding on GPIbα, leading to its degradation [27]. In addition, antibody-mediated platelet destruction occurs via Fc receptors on primarily splenic macrophages and it is frequent in primary immune thrombopenia (ITP) [28]. In this disease, circulating autoantibodies with specificity for membrane glycoproteins, such as GPIIb-IIIa or GPIbα, can bind to platelets, thus triggering platelet desialylation by secretion of active Neu1, and additionally favoring their clearance by cytotoxic CD8 T lymphocytes [29,30].

Platelet survival also depends on the interplay between antiapoptotic and proapoptotic factors of the Bcl-2 family, which are critical regulators of the intrinsic apoptotic pathway [31] (Figure 2B). However, it is still unclear whether Bcl-2 family members alter the sialic acid content on the surface of platelets. Platelet loss of function and death is governed by unclear mechanisms that share some similarity to those used by nucleated cells for programmed cell death [32]. In addition, platelets express certain components of the extrinsic pathway of apoptosis, including caspase 8, but the limited data available to date do not support their critical role in regulating platelet lifespan [33]. The consequences of platelet death include the formation of a new platelet–platelet interaction that occurs between nonviable platelets, and the shedding of the collagen receptor GPVI and GPIbα. Both processes appear to be regulated by metalloproteinase activity [34]. Although it is unclear how senescent platelets are removed from circulation, many cells undergoing apoptosis shift the redistribution of phosphatidylserine (PS) from the inner to the outer lamella of the plasma membrane, which serves as a molecular signal for removal by phagocytes [35] (Figure 2B). Overall, it remains to be elucidated whether loss of sialic acid triggers the intrinsic apoptotic machinery in platelets during the clearance mechanisms that regulate platelet counts.

Thousands of enzymes regulated by glycosylation processes are involved in platelet formation and clearance. Alterations in any of them could result in an imbalance between the two processes and consequently impact platelet counts. Until relatively recently, a very limited number of molecular variants had been described in only few genes that were related to IT [36].

## 3. Disorders of Glycosylation Associate with Syndromic Thrombocytopenia

Congenital disorders of glycosylation (CDG) include a rapidly growing group of metabolic diseases that are caused by molecular defects in genes involved in glycoprotein synthesis. To date, more than 100 types of CDGs has been described [37,38]. These inherited disorders are associated with a wide variety of multiorgan symptoms, although the molecular alterations associated with IT and/or other hematologic manifestations involve a small number of genes that have been described recently [10].

### 3.1. Disorders of Glycosylation Described in Patients with Thrombocytopenia

#### 3.1.1. GNE-Related Disorder

The first documented evidence of IT associated with a molecular alteration in an enzyme involved in glycosylation occurred in 2014; Izumi et al. reported two siblings with myopathy, rimmed vacuoles, and inherited thrombocytopenia harboring two compound heterozygous *GNE* mutations, p.Val603Leu and p.Gly739Ser, in accordance with autosomal recessive inheritance of the disease. The authors speculated that decreased GNE activity would lead to decreased sialic content in platelets [39], as the *GNE* encodes for UDP-N-acetylglucosamine 2-epimerase, a bifunctional enzyme that catalyzes the initial two steps in sialic acid biosynthesis and regulates total levels of N-acetylneuraminic acid, a precursor of sialic acids [40] (Figure 3). In the same year, Zhen et al. reported two adult siblings with thrombocytopenia and compound heterozygous *GNE* mutations (p.Tyr217His and p.Asp515Glnfs*2). These patients showed mild to moderate thrombocytopenia and no overt bleeding [41].

The GNE-related disorder was expanded in 2018, with the publication and characterization of several unrelated pedigrees [42,43]. One of the pedigrees was an inbred family carrying GNE p.Gly416Arg in homozygosis, in which the patients had severe macrothrombocytopenia with a high immature platelet fraction [42]. Similarly, Revel-Vilk et al. reported nine affected individuals from three unrelated families with severe macrothrombocytopenia, bleeding tendency, and a high proportion of reticulated and desialylated platelets [43]. Of note, none of the patients in the different pedigrees had myopathy. These studies suggest that several mechanisms of platelet clearance and production may be affected by desialylation. Patients have a rapid platelet clearance associated with loss of the platelet surface GPIb/IX receptors and changes in surface sialylation, suggesting a strong link between sialylation, altered surface GPIb/IX, increased platelet size, and platelet clearance [42,43]. However, it is still unclear whether platelets from patients with *GNE*-related disorder without sialic acid are cleared from the circulation by AMR, nor is it known whether variants in *GNE* are associated with alterations in MK maturation and platelet formation. It has been hypothesized that mutations in *GNE* cause thrombocytopenia only when co-segregated with other genetic factors, such as *ANKRD18A, FRMPD1, FLNB,* and *PRKACG*, which have been described in other cases [44].

Recently, new patients carrying biallelic variants of *GNE* have been published [45,46], however, it is still unclear why some patients present with isolated thrombocytopenia while others present with myopathy. Considering that GNE-related myopathy usually appears in the third decade of life, we cannot exclude that patients presenting with only thrombocytopenia develop myopathy later in life [40]. Further studies are needed to better understand why variants in *GNE* are associated with three distinct clinical phenotypes: myopathy, sialuria, or isolated thrombocytopenia. 

#### 3.1.2. SLC35A1-Related Disorder

In 2011, the biallelic genetic mutation in the *SLC35A1* gene, which encodes the cytidine-5′-monophosphate [CMP]-sialic acid transporter that transfers CMP sialic acid from the nucleus to the Golgi apparatus for sialylation, was described for the first time (Figure 3). Sialyltransferases constitute a family of glycosyltransferases that transfer sialic acid from the donor substrate to acceptor oligosaccharide substrates. Thus, impaired transporter function results in a defect of α2,3-sialylation, causing thrombocytopenia in patients due to decreased platelet sialylation and increased clearance [47]. In addition, the authors demonstrated the presence of giant platelets with morphological abnormalities, such as open canalicular membrane system of platelets, and showed an increased number of small MKs. These results suggest defective megakaryopoiesis based on hyposialylation that may interfere with membrane-forming processes. However, in 2018, Kauskot et al. reported the congenital deficiency in SLC35A1 in two siblings born to consanguineous parents, who presented with delayed psychomotor development, epilepsy, ataxia, microcephaly, choreiform movements, and mild macrothrombocytopenia. In fact, they had a high proportion of immature platelets, suggesting that platelet formation may also be impaired, as previously reported. The authors speculated that SLC35A1 is relevant for platelet life span but not for proplatelet formation, and that the giant platelets could correspond to a compensatory mechanism in a context of thrombocytopenia, as suggested by elevated levels of reticulated platelets and an increased MK count in the bone marrow [48].

Recently, Ma et al. provided new insights into the role of sialylation in platelet homeostasis and the mechanisms of thrombocytopenia in SLC35A1-related disorder by generating a mouse model of the disease. They demonstrated that the number of bone marrow MK in *Slc35a1*-/- mice was reduced, and their maturation was also impaired. In addition, the authors reported an increased number of desialylated platelets that were removed by Küpffer cells in the liver of *Slc35a1*-/- mice [49]. 

Overall, further studies are needed to demonstrate the exact role of SLC35A1 in megakaryopoiesis. Although thrombocytopenia is known to be associated with increased clearance, the mechanisms are still unclear, and there is great controversy about its role in MK maturation and proplatelet formation. Further studies in new patients are mandatory in order to clarify the discrepancies detected between the patient described by Kauskot et al. [48], and the animal model generated by Ma and colleagues [49].

#### 3.1.3. GALE-Related Disorder

The *GALE* gene encodes uridine diphosphate [UDP]-galactose-4-epimerase, which catalyzes the bidirectional interconversion of UDP-glucose to UDP-galactose, and of UDP-N-acetyl-glucosamine to UDP-N-acetyl-galactosamine (Figure 3). Thus, GALE balances, by reversible epimerization, the pool of four sugars that are essential during the biosynthesis of glycoproteins and glycolipids [50,51].

The first evidence of UDP-galactose-4-epimerase deficiency associated with hematological alterations was reported in 1995, in a four-year-old girl presenting with bruising, thrombocytopenia, and dysplastic cells in the bone marrow. However, the molecular diagnosis was not performed, and the underlying *GALE* variants are unknown [52].

In 2019, Seo et al. reported six members of a consanguineous family carrying the GALE variant p.Arg51Trp in homozygosis, all affected by anemia, febrile neutropenia, and severe thrombocytopenia, associating increased hemorrhagic tendency, without symptoms of systemic galactosemia [53], providing the first evidence of *GALE* variants and hematologic alterations. In 2020, Febres-Aldana et al. described a child with bone marrow dysfunction and complex congenital heart disease associated with compound heterozygosity in GALE (p.Arg51Trp and p.Gly237Asp) [54]. In addition, in 2021, Markovitz et al. reported a patient with pancytopenia and immune dysregulation due to the previously described homozygous p.Thr150Met variant of GALE [55]. Although three pedigrees carrying GALE variants associated with hematological abnormalities and different phenotypes had been reported, there was no evidence on the mechanism leading to disease in patients carrying GALE variants. 

In 2022, we unveiled four *GALE* variants associate with reduced glycosylation of GPIbα and β1 integrin causing impaired externalization to the surface of MK and platelets, altering the distribution of F-actin and filamin A in MKs, and affecting platelet production. In addition, hypoglycosylated and non-functional platelets prone to apoptosis were observed. Overall, these findings demonstrated the essential role of GALE in glycosylation, platelet formation, function and clearance, providing new clues to understand the biological mechanisms underlying the biology and pathophysiology of the β1 integrin and GPIb-IX-V complex [22]. 

Notwithstanding, the nature and severity of symptoms in epimerase deficiency remain unclear, as do the mechanisms by which some variants are associated with severe syndromic disorders that include hematological manifestations, while others are not [56]. Extending the analysis to additional receptors or other crucial glycoproteins may open new avenues toward understanding the impact of glycosylation on megakaryopoiesis. In addition, further studies are needed to provide new insights into the mechanisms associated with platelet clearance to elucidate a possible link between hypoglycosylated platelets, clearance by AMR or Kupper cells, and mechanisms of apoptosis.

#### 3.1.4. β4GALT1-Related Disorder

The β-1,4-galactosyltransferase 1 (β4GALT1) is an enzyme that transfers galactoses from UDP-Gal to terminal N-acetylglucosamine (GlcNAc) (Figure 3). To date, only a few cases of inherited disorders of glycosylation by β4GALT1 have been described. Until 2020, these comprised three patients, all with clinical features including hypotonia, coagulopathy, elevated serum transaminases and a type 2 biochemical pattern on serum transferrin isoform analysis [57,58,59]. Staretz-Chacham et al. described three additional patients homozygous for a novel mutation in β4GALT1 (p.Arg21Trp), located within its transmembrane domain. These patients showed a uniform clinical presentation with intellectual disability, profound pancytopenia requiring chronic treatment, and novel features including pulmonary hypertension and nephrotic syndrome [60]. In addition, Giannini et al. generated a *B4galt1*-/- mouse and observed that β4GALT1 deficiency increases the number of differentiated MKs. The resulting lack of glycosylation potentiates β1 integrin signaling, resulting in the differentiation of dysplastic MKs with severe alterations in the formation of the demarcation system and thrombopoiesis. Impaired thrombopoiesis also led to increased plasma TPO levels and defective hematopoietic stem cells (HSCs), justifying the observed thrombocytopenia [61]. These finding were in agreement with those published by Di Buduo et al. who reported increased *B4GALT1* gene expression and plasma TPO levels in patients with myeloproliferative neoplasms (MPNs) [62]. Here, the altered B4GALT1 expression in MPN MKs led to the production of platelets with aberrant galactosylation, which in turn promoted hepatic TPO synthesis independently of platelet count [62]. 

The characterization of a larger number of patients with β4GALT1 deficiency is required for a better understanding of the pathophysiological mechanisms underlying the disease, and to establish a correlation between the molecular alteration and the disease manifestations.

#### 3.1.5. Other CDGs with Potential Relation to Inherited Thrombocytopenia in Patients

ALG1-CDG: This autosomal recessive disorder is caused by the deficiency of the 1,4-mannosyltransferase 1 enzyme, encoded by *ALG1* gene. Patients commonly suffer from severe neurological manifestations, developmental and psychomotor delay, with variable affectation of other organs (nephrotic syndrome, ascites, hepatomegaly, cardiomyopathy, ocular manifestations, and immunodeficiency). Hematological abnormalities, including thrombocytopenia, were found in approximately 50% of the patients, but detailed platelet analyses have not been reported yet [63].

ALG8-CDG: The *ALG8* encodes the α-1,3-glucosyltransferase. The dysfunction of the enzyme leads to a severe disease characterized by gastrointestinal and cognitive impairment, edema, and dysmorphism, resulting in the death of patients within the first year of life. In addition, most patients presented thrombocytopenia, but mechanisms were not characterized [64].

MPI-CDG: The mannose phosphate isomerase (MPI) is involved the first step of the GDP-mannose synthesis (i.e., the conversion of fructose-6-phosphate to mannose-6-phosphate). It plays a critical role in maintaining the supply of D-mannose derivatives required for most glycosylation reactions [65]. MPI-CDG does not cause as significant neurologic and multi-systemic involvement, but patients show a hepatic–intestinal presentation comprising life-threatening gastrointestinal bleeding. Pancytopenia, including moderate thrombocytopenia, has been reported in one adult, but it is still essential to confirm the role of MPI in thrombopoiesis to rule out a different etiology for the inherited thrombocytopenia [66].

PMM2-CDG: the phosphomannomutase 2 (PMM2) catalyzes the isomerization of mannose 6-phosphate to mannose 1-phosphate, which is subsequently converted into GDP-mannose (the source of mannose for the glycosylation branches). It is by far the most common N-glycosylation disorder. Biallelic pathogenic variants associate with a multisystem disease with highly variable phenotype. In the infantile multisystem presentation, infants show axial hypotonia, hyporeflexia, esotropia, and developmental delay. During late-infantile and childhood, they display ataxia–intellectual disability stage (ataxia, severely delayed language and motor development, inability to walk, among others). In the adult stable, the peripheral neuropathy is variable, and it is common to diagnose progressive retinitis pigmentosa and myopia, thoracic and spinal deformities with osteoporosis worsen, and premature aging. Moreover, females may lack secondary sexual development and males may exhibit decreased testicular volume [67]. Despite the increased risk to deep venous thrombosis is a common characteristic of the disease, patients with unusual thrombocytopenia have also been reported [68,69]. However, additional studies are required to establish the causality of thrombocytopenia and thrombosis in these patients. 

MAGT1-CDG: The magnesium transporter 1 (MAGT1) critically mediates magnesium homeostasis. Its alteration results in X-linked immunodeficiency, thus, most patients developed chronic EBV-associated B cell lymphomas, caused by the altered homeostasis in T-helper, cytotoxic T-lymphocytes, and natural killer cells. Moreover, these patients present with a phenotype that is mainly characterized by intellectual and developmental disability [70]. Some patients develop mild to moderate thrombocytopenia, although the mechanisms of pathogenicity affecting megakaryocytes and platelets have not been reported. Magt^–/y^ mice have normal platelet count and size but altered ploidy of megakaryocytes [71]. It is important to mentioned that, in MAGT1-deficient cells, Mg^2+^ supplementation increased the free intracellular Mg^2+^ levels, most likely through *TRPM7* (Transient receptor potential cation channel subfamily M member 7), which molecular alteration has been related to thrombocytopenia [72]. Therefore, it is necessary to further investigate the role of *MAGT1* in platelet formation and its association with *TRPM7*.

### 3.2. Disorders of Glycosylation Associated to Syndromic Thrombocytopenia Reported Only in Mice Models

#### 3.2.1. ST3GAL4-Related Disorder

The gene *ST3GAL4* codifies for the ST3Gal-IV enzyme, a sialyltransferases that transfers the sialic acid in α2,3 linkage to the acceptor oligosaccharide substrates, i.e., glycans with terminal Galβ1-4GlcNAc, Galβ1-3GlcNAc, and Galβ1-3GalNAc sequence. A recent study published in 2022 by Wiertelak W. and colleagues, demonstrate that ST3GAL4 associate with SLC35A1 forming a complex essential for N-glycan α2,3 sialylation [73]. Therefore, it is expected that molecular alterations in *ST3GAL4* are associated with a phenotype similar to that observed in patients with SLC35A1-RD, where platelets have an increased clearance from bloodstream, leading to thrombocytopenia.

Moreover, the investigations performed on knock-out (KO) mice for ST3Gal-IV (ST3Gal-IV^−/−^) demonstrate that platelets were removed rapidly from circulation, and that biphasic kinetics was followed by a fast initial clearance and a prolonged clearance phase [23]. These ST3Gal-IV^−/−^ platelets were removed in the liver by asialoglycoprotein receptors on macrophages and hepatocytes. Among the major desialylated proteins in ST3Gal-IV^−/−^ lysates, authors revealed the presence of GPIba with increased exposure of βGlcNAc residues (thus, desialylated platelets). Finally, authors revealed that megakaryopoiesis was not increased in ST3Gal-IV^−/−^ mice despite accelerated platelet clearance [23]. These results are in accordance with those observed in *Slc35a1* KO mice [49]. In addition, Qi F, et al. revealed that the α2,3-sialylation levels of β1 integrin were clearly suppressed in the ST3GAL4 KO cells lines, supporting another target of molecular defects in genes involved in congenital disorders of glycosylation [74].

However, no patients with alterations in *ST3GAL4* have been reported to date, so it cannot be ruled out that patients may have a defect in thrombopoiesis, as these same discrepancies between humans and mice remain unresolved for alterations in *SLC35A1*.

#### 3.2.2. ST3GAL1-Related Disorder

ST3GAL1, encoded by *ST3GAL1* gene, is a sialyltransferase that transfer sialic acid to the galactose residue of type III disaccharides (Galβ1,3GalNAc). The conditional KO mice model in the MK lineage (St3gal1^MK−/−)^ displayed a 50% reduction in platelet counts vs. control, with increased mean platelet volume (MPV) and immature platelet fraction (IPF). Erythrocytes and leukocytes counts were normal. Moreover, St3gal1^MK−/−^ platelet life span and expression of the platelet surface receptors glycoprotein IIb (GPIIb), GPIIIa, GPIbα, GPIX, GPV, and GPVI were comparable to controls, in contrast to alterations in the *GNE, GALE* or *ST3GAL4* genes, where we detected reduced levels of GPIbα and/or β1 integrin. Lastly, transfused ST3Gal1^MK−/−^ platelets were not recognized by the AMR, as evidenced by similar survival in WT, and hepatic TPO production was also indistinguishable between ST3Gal1^MK−/−^ mouse and control livers [75].

Conversely, recent research revealed that both ST3GAL1 and ST3GAL2 became highly expressed during the differentiation of human-induced pluripotent stem cells (iPSCs) into hematopoietic progenitor cells (HPCs), but their expression decreased markedly upon differentiation into MKs. Interestingly, the HPC markers CD34 and CD43, as well as the MK membrane marker GPIbα, were identified as major GP substrates for ST3GAL1 [76], contrary to what has been described in the animal model ST3Gal1^MK−/−^ [75]. The authors concluded that disruption of ST3GAL1 had little impact on MK production, but its absence resulted in dramatically impaired MK proplatelet formation [76].

#### 3.2.3. C1GALT1-Related Disorder

Core 1 β1,3-galactosyltransferase (C1GalT1) catalyzes the formation of core 1 O-glycan structures, a common precursor for mucin-type O-glycans. Impaired C1GalT1 activity has been associated with different disorders in humans, such as the Tn syndrome (a rare autoimmune disease in which subpopulations of blood cells in all lineages carry an incompletely glycosylated membrane) and IgA nephropathy, a common primary glomerulonephritis [77,78].

The murine model expressing very low residual enzymatic activity (C1GalT1 mice) revealed a 40% reduction of platelet counts compared to WT mice and increased platelet volume. Other blood cells counts were unaffected. There was no reduction in megakaryocyte numbers and DNA ploidy, and the electron microscopic evaluation of MKs and platelets from C1GalT1 mice vs. WT suggested no major obvious ultrastructural abnormalities. Moreover, the half-life of platelets in C1GalT1 mice was similar to control mice, but the generation of unlabeled platelets after pulse labeling occurred at a slower rate. Thus, authors suggested that the thrombocytopenia in C1GalT1 mice is not caused by impaired megakaryocyte production or accelerated clearance of platelets but seems to be caused by compromised thrombopoiesis [79].

In accordance, Kudo T and colleagues exploited an interferon-inducible Mx1-Cre transgene to conditionally ablate the C1galt(flox) allele (Mx1-C1). Mx1-C1 mice exhibit severe thrombocytopenia, giant platelets, and prolonged bleeding times. Both the number and DNA ploidy of megakaryocytes in Mx1-C1 bone marrow were normal. However, they found very few proplatelets in Mx1-C1 primary megakaryocytes. Protein levels revealed a reduced expression of GPIbα in Mx1-C1 mice and circulating Mx1-C1 platelets exhibited an increase in the number of microtubule coils, despite normal levels of α- and β-tubulin [80]. 

Results in both mice models of C1GalT1 deficiency demonstrate that O-glycan is required for terminal megakaryocyte differentiation and platelet production. Considering that the biological importance of O-glycans in platelet clearance was unclear, Li Y and colleagues generated mice with a hematopoietic cell-specific loss of O-glycans (HC C1galt1-/-). These mice also exhibit reduced peripheral platelet numbers with reduced levels of α-2,3-linked sialic acids and increased platelet accumulation in the liver compared to WT platelets, demonstrating that hepatic AMR promotes preferential adherence and phagocytosis of desialylated platelets by the Kupffer cell through its C-type lectin receptor CLEC4F [81].

#### 3.2.4. COSMC-Related Disorder 

The core-1 β1-3galactosyltransferase-specific chaperone 1 (Cosmc) is an essential chaperone that functions in the endoplasmic reticulum (ER) regulating protein O-glycosylation and helping C1GALT1 to fold correctly [82].

Cosmc-KO mice exhibit embryonic lethality, while the inducible CAGCre-ERTM/Cosmc-KO (iCAG-Cos) mice exhibited a global loss of core 1-derived O-glycans, high mortality, leukocytopenia, thrombocytopenia, severe acute pancreatitis, atrophy of white and brown adipose tissue, spontaneous gastric ulcers, and severe renal dysfunction [83]. Moreover, targeted deletion of Cosmc in murine endothelial/hematopoietic cells (EHC) (EHC Cosmc(-/y)) showed that platelets exhibited a marked decrease in GPIb-IX-V function and agonist-mediated integrin αIIbβ3 activation, associated with loss of interactions with von Willebrand factor and fibrinogen, respectively [19]. 

## 4. Conclusions and Perspectives

Glycoconjugates are major components of animal cells with an essential role in many physiological processes. Advances in glycobiology and the development of mass spectrometry-based proteomics and glycomics have uncovered the mechanism of aberrant glycosylation in a wide spectrum of congenital disorders and elucidated the functions of specific glycans and related genes [84,85]. In recent years, these approaches have led to the discovery of novel genes involved in different pathologies. In the field of hematology, no gene involved in glycosylation affecting megakaryopoiesis was known until 2014 [39]. To date, only alterations in the *GNE, SLC35A1, GALE* and *B4GALT1* genes causing inherited thrombocytopenias have been described and probed in patients. However, the mechanism of thrombocytopenia and platelet clearance associated with variants in these genes remains to be fully elucidated. Studies in both human patients and animal models of the disease reveal that altered N- and O-glycosylation of essential platelet proteins such as GPIbα and the β1 integrin underlie the mechanism of pathogenicity causing an increased platelet clearance, mainly mediated by the liver, and abnormal thrombopoiesis with no remarkable changes in megakaryocyte maturation. Further research of these mechanisms is essential, as well as to understand why not all patients carrying biallelic mutations in these genes develop thrombocytopenia and severe syndromic manifestations. The emerging increase over the last few years in the study of glycosylation disorders is allowing the discovery of novel genes involved in platelet formation and function. So far, alterations in genes such as *ST3GAL4*, *ST3GAL1*, *C1GALT1* or *COSMC* have only been reported in animal models, but it is expected that in the coming years, and with the rise of high-throughput sequencing techniques, patients with these alterations will be reported.

The discovery of aberrant glycans and exploration of the underlying mechanisms would broaden their applications as diagnostic markers or therapeutic targets, improving patient care. It is important to mention that disorders of glycosylation affect people from birth, though symptoms may manifest later. Considering the serious syndromic manifestations, an accurate and early diagnosis is essential for treatment of these patients. TPO receptor agonist could be an alternative to platelet transfusion as described in other Its [86] and, in selected severe patients, the hematopoietic stem cell transplantation (HSCT) may be an option. A recent case study was published documenting the first HSCT in a patient with an inherited defect of *GNE* resulting in a normal platelet count [87], raising the horizon in the field of congenital disorders of glycosylation. Finally, gene therapy may be a promising approach for the future of these patients by ex vivo correction of variants detected in patients by the wild-type form of the protein.

## Figures and Tables

**Figure 1 ijms-24-05109-f001:**
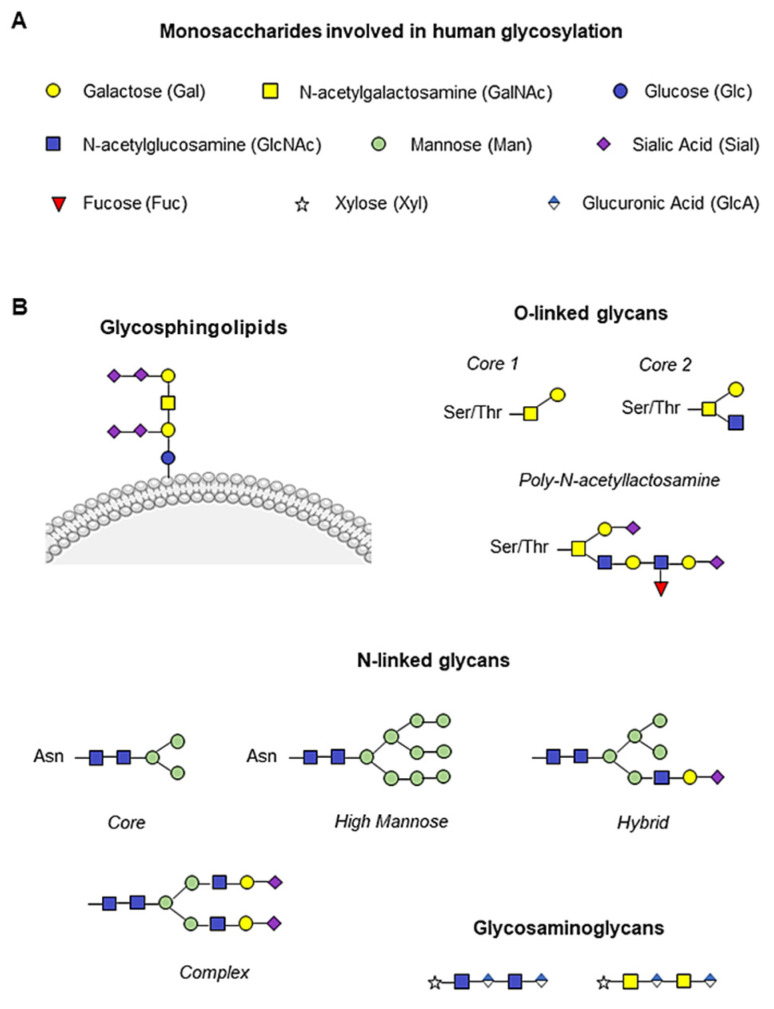
Human monosaccharides and their glycoconjugate forms. (**A**) Representation of the nine monosaccharides involved in human glycosylation. (**B**) Schematic illustration of glycan structures. Addition of N-acetylgalactosamine (GalNAc, yellow square) to serine/threonine (Ser/Thr) residues initiates O-glycan synthesis, while two N-acetylglucosamine (GlcNAc, blue square) and three mannose (Man, green circle) constitute the N-glycan core, and the glycosylation branches are formed by galactose molecules (Gal, yellow circle), GalNAc, GlcNAc, fucose (Fuc, red triangle) and the final sialic acid (Sial, purple rhombus). Glycosphingolipids are conjugated to the plasma membrane, whereas glycosaminoglycans are mainly composed of an initial xylose (Xyl, white star) followed by glucuronic acid (GlcA, blue and white rhombus) and GlcNAc or GalNAc.

**Figure 2 ijms-24-05109-f002:**
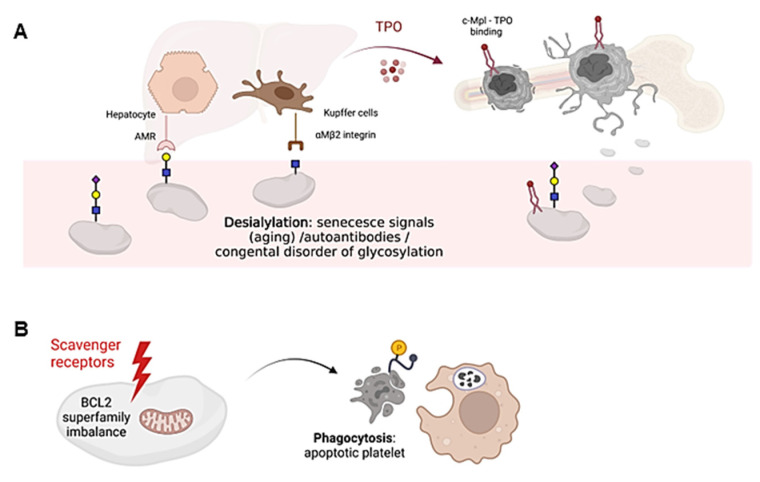
Schematic representation of platelet clearance mechanisms. (**A**) Platelet desialylation can be triggered by senescence signals in “aged” platelets, by the presence of autoantibodies (especially in autoimmune diseases) or by congenital alterations in genes involved in glycosylation. Desialylated platelets are recognized by hepatic AMR to regulate hepatic TPO production and thrombopoiesis, or by Kupffer cells. Bone marrow MKs produce and release young platelets containing sialic acid into the bloodstream. Young platelets maximally internalize TPO through Mpl receptors. (**B**) Platelet survival is regulated by the interaction between pro-survival and pro-apoptotic members of the Bcl-2 family, which are critical regulators of the intrinsic apoptotic pathway. Up-regulation of pro-apoptotic signals mediates platelet clearance through scavenger receptors. Cells undergoing apoptosis modify the redistribution of phosphatidylserine (PS, yellow circle) from the inner to the outer lamella of the plasma membrane, which serves as a molecular signal for clearance by phagocytes. The figure was created Biorender.com (accessed on 13 February 2023).

**Figure 3 ijms-24-05109-f003:**
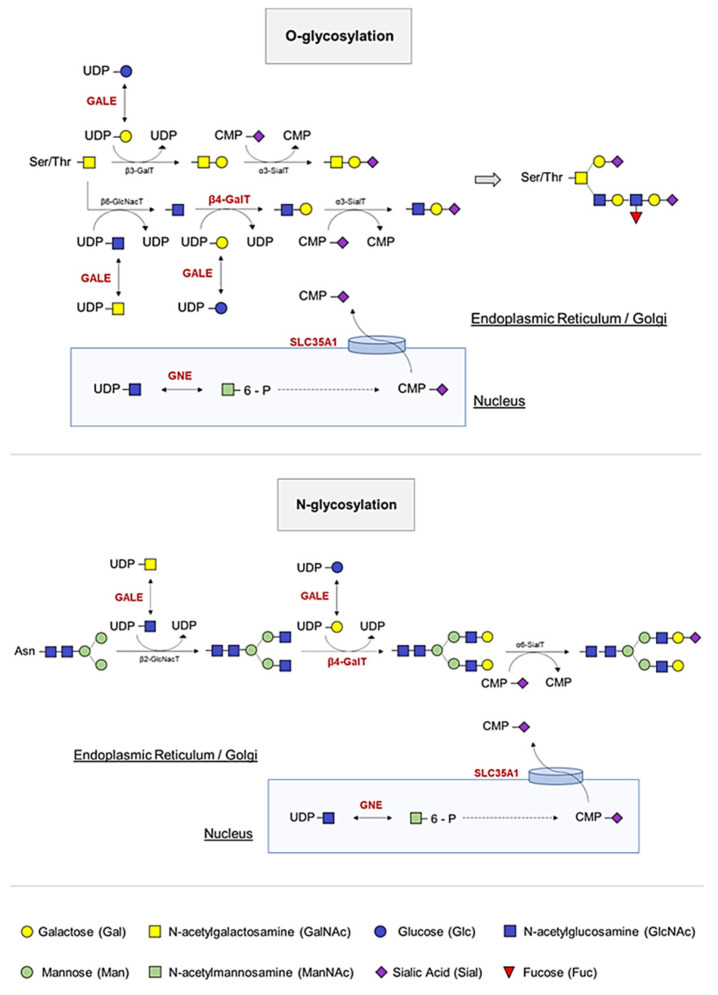
O- and N-linked glycosylation and sialylation of platelets. The GALE protein allows the interconversion of four essential molecules in the glycosylation process by serving as substrates for other enzymes, which incorporate the carbohydrates of interest and release UDP. During glycosylation, branching of carbohydrates occurs. B4GALT1 transfers galactoses from UDP-Gal to the terminal GlcNAc. A final cleavage of sialic acid prevents platelet clearance. GNE and SLC35A1 participate in the pathway that allows incorporation of this sialic acid into platelet proteins.

## Data Availability

Not applicable.

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
