# Peer review of "Inherited Thrombocytopenia Caused by Variants in Crucial Genes for Glycosylation"

_ijms, 2023, doi:10.3390/ijms24065109_

Round 1

Reviewer 1 Report

This is an exciting research paper.

However, a few suggestions are placed that need to be clarified or included to further improve the manuscript.

Comment 1: Are these glycosylation defects associated with Inherited Thrombocytopenia alone or there are syndromes?

Comment 2: Do they all present at same age or there are diversity in presentation?

Comment 3: Are there any extra-haematological manifestations which are characteristic of glycosylation defects and harbinger thrombocytopenia.

Comment 3: The therapeutic options/management can be described a bit more.

Comment 4: Are there difference in phenotype (severity of disease) of patients having glycosylation defects? What are the factors deciding the phenotype?

Comment 5: Are there going to be any changes in the existing clinical practice of managing Inherited Thrombocytopenia?

Author Response

Reviewer 1

This is an exciting research paper. However, a few suggestions are placed that need to be clarified or included to further improve the manuscript.

We thank the Reviewer for the positive feedback and helpful comments.

Comment 1: Are these glycosylation defects associated with Inherited Thrombocytopenia alone or there are syndromes?

We thank the reviewer for the helpful observation. Defects in GNE, SLC35A1, GALE and B4GALT are associated with syndromic manifestations, including inherited thrombocytopenia. The common syndromic manifestations of patients carrying GNE biallelic variants is myopathy (page 6, line 2). However, several patients with GNE and thrombocytopenia without myopathy have been described (page 6, second paragraph). Regarding congenital deficiency in SLC35A1, patients reported by Kauskot et al.  in 2018, showed clinical presentations including delayed psychomotor development, epilepsy, ataxia, microcephaly, choreiform movements, and mild macrothrombocytopenia (page 6, last paragraph). GALE related disorder is associated with a wide range of syndromic manifestations, that may include learning difficulties, delayed growth, sensorineural hearing loss, and early-onset cataracts and, less frequently, cardiac failure and hepatomegaly (Marín-Quílez A, et al. Platelets 2023; Marín-Quílez A, et al. Blood 2022). Finally, patients with β4GALT1 variants had been reported with intellectual disability, profound pancytopenia requiring chronic treatment, pulmonary hypertension, and nephrotic syndrome (page 8, lines 7-14).

Comment 2: Do they all present at same age or there are diversity in presentation?

We greatly appreciate this suggestion. Congenital disorders of glycosylation (CDG) affect patients since birth. In some cases, they are diagnosed and treated since childhood, but in other cases, such as the patients with GALE variant recently published by our group (Marin-Quilez A, et al. Blood 2022), they have been diagnosed at the age of 40, since CDG are rare, and underestimate disorders whose role in human hematology are poorly known. We included a brief commentary on page 10, second paragraph.

Comment 3: Are there any extra-haematological manifestations which are characteristic of glycosylation defects and harbinger thrombocytopenia.

We thank the reviewer for the interesting question. Patients with congenital disorders of glycosylation presenting with thrombocytopenia display a wide range of syndromic manifestations, as described in comment 1. To the best of our knowledge, there is not a characteristic extra-hematological manifestation. Several GNE patients had been diagnosed with myopathy, and few patients with GALE variants suffered from cardiopathy, however, the feature is not considered as characteristic of the disorder. It is important to note that this kind of disorders are congenital and hereditary, so that all organs have the enzyme deficiency and patients could manifest different symptoms.

Comment 3: The therapeutic options/management can be described a bit more.

 As suggested by the reviewer, we deeply described the therapeutic options for patients with congenital disorders of glycosylation and thrombocytopenia: “Considering the serious syndromic manifestations that patients present, an accurate and early diagnosis is essential for their treatment. TPO analogues is a promising therapy to rescue the severe thrombocytopenia that patients with GNE, SLC35A1, GALE, and B4GALT1 present as an alternative to blood transfusions. However, the only definitive treatment offered to date to these patients, and in selected cases, is the hematopoietic stem cell transplantation (HSCT). Recently, it has been published the first HSCT in a patient with an inherited defect of GNE, resulting in a normal platelet count [65], and raising the horizon in the field of congenital disorders of glycosylation. Finally, gene therapy may be a promising approach for the future of this patients by ex vivo correction of variants detected in patients by the wild-type form of the protein.” (Page 10, second paragraph).

Comment 4: Are there difference in phenotype (severity of disease) of patients having glycosylation defects? What are the factors deciding the phenotype?

We would like to express our acknowledgment to this Reviewer for the positive feedback.

We would like to express our acknowledgment to the reviewer for pointing out this interesting observation. Patients may develop a wide range of clinical manifestations, from "benign" symptoms such as galactosemia in patients with some GALE variants (Marín-Quilez A, Platelets 2023), to life-threatening and syndromic phenotypes affecting more than four or five different organs. As we mentioned on page 10, lines 7-8, we do not still understand why only a few of the patients carrying biallelic mutations in these genes develop thrombocytopenia and severe syndromic manifestations. Probably the phenotype is associated with the type of variant and its structural change. However, the reduced number of patients described is not enough to stablish link and it still needs to be further elucidated.

Comment 5: Are there going to be any changes in the existing clinical practice of managing Inherited Thrombocytopenia?

This is a very interesting question. At this moment, and considering our experience with patients carrying GALE variants, it should be noted that since the molecular diagnosis was reached and patients were notified that they suffered from GALE-RD, they have a greater clinical follow-up by different specialists to prevent the appearance of symptoms. For example, GALE-RD is associated with premature cataracts, so patients have ophthalmologic surveillance to prevention and early intervention. Many times, these patients are diagnosed late, due to the lack of knowledge about these rare diseases, so an early diagnosis is expected to control the symptoms and prevent them from worsening. As for therapies, only one bone marrow transplant has been performed in a patient with GNE, but it may be a therapy in the near future for patients with alterations in the other genes.

Reviewer 2 Report

I found this review very well presented, written and informative. It clearly shows the complexity of glycosylation and questions its importance in inherited thrombocytopenia.

A minor point.

While figure 3 shows the process of glycolysation and indicates the different enzymes involved, it would be even more instructive to see the result of variant defects on glycosylation.

The triangle symbolising Fucose is oriented upwards in the legend and downwards in figures 1 and 3.

Author Response

Reviewer 2

I found this review very well presented, written and informative. It clearly shows the complexity of glycosylation and questions its importance in inherited thrombocytopenia.

We would like to express our acknowledgment to this Reviewer for the positive feedback.

A minor point. While figure 3 shows the process of glycolysation and indicates the different enzymes involved, it would be even more instructive to see the result of variant defects on glycosylation.

We thank the reviewer for the helpfully observation. We agree that showing glycosylation defects caused by variants in genes would be very instructive. However, it is not known exactly how the addition of sugars is altered.

In the case of the GNE and SLC35A1variants it is known that the final sialic acid is not added, altering the sialylation of the proteins, and thus favoring platelet clearance. In the case of b4galt1, it has been shown that the final galactose is not added, so it is expected that glycosylation is altered at that level. However, in the case of the GALE variants, and considering that it is an enzyme that acts reversibly, it is not known whether its dysfunction results in a defect during the addition of GluNAc or Gal to the branches. It has also not been demonstrated whether there is a lower exposure of sialic acid in the proteins by lectin binding studies, as this is something we want to do soon with our patients with GALE variants, so it is too early to establish the exact point where glycosylation is truncated.

The triangle symbolising Fucose is oriented upwards in the legend and downwards in figures 1 and 3.

We thank the reviewer for the suggestion, and we change the legend in Figure 1 and 3.

Round 2

Reviewer 1 Report

Nicely modified

Author Response

Thank you for your helpful comments and for improving the quality of the work.